# Specificity of end resection pathways for double-strand break regions containing ribonucleotides and base lesions

James M. Daley [1,2✉], Nozomi Tomimatsu[3], Grace Hooks[1,6], Weibin Wang[1,7], Adam S. Miller[1,8], Xiaoyu Xue[1,4], Kevin A. Nguyen[1,9], Hardeep Kaur[2], Elizabeth Williamson[5], Bipasha Mukherjee[3], Robert Hromas[5], Sandeep Burma [2,3✉] & Patrick Sung [1,2✉]

DNA double-strand break repair by homologous recombination begins with nucleolytic resection of the 5′ DNA strand at the break ends. Long-range resection is catalyzed by EXO1 and BLM-DNA2, which likely have to navigate through ribonucleotides and damaged bases. Here, we show that a short stretch of ribonucleotides at the 5′ terminus stimulates resection by EXO1. Ribonucleotides within a 5′ flap are resistant to cleavage by DNA2, and extended RNA:DNA hybrids inhibit both strand separation by BLM and resection by EXO1. Moreover, 8-oxo-guanine impedes EXO1 but enhances resection by BLM-DNA2, and an apurinic/apyrimidinic site stimulates resection by BLM-DNA2 and DNA strand unwinding by BLM. Accordingly, depletion of OGG1 or APE1 leads to greater dependence of DNA resection on DNA2. Importantly, RNase H2A deficiency impairs resection overall, which we attribute to the accumulation of long RNA:DNA hybrids at DNA ends. Our results help explain why eukaryotic cells possess multiple resection nucleases.

[1] Department of Molecular Biophysics and Biochemistry, Yale University School of Medicine, New Haven, CT 06510, USA. [2] Department of Biochemistry and Structural Biology, University of Texas Health Science Center, San Antonio, TX 78229, USA. [3] Department of Neurosurgery, University of Texas Health Science Center, San Antonio, TX 78229, USA. [4] Department of Chemistry and Biochemistry, Texas State University, San Marcos, TX, USA. [5] Department of Medicine, University of Texas Health Science Center, San Antonio, TX 78229, USA. [6]Present address: Department of Biochemistry, Duke University, Durham, NC 27710, USA. [7]Present address: Department of Radiation Medicine, School of Basic Medical Sciences, Peking University Health Science Center, Beijing 100191, China. [8]Present address: Regeneron, Rensselaer, NY 12144, USA. [9]Present address: David Geffen School of Medicine, University of California Los Angeles, Los Angeles, CA 90095, USA. ✉email: daleyj@uthscsa.edu; burma@uthscsa.edu; sungp@uthscsa.edu

DNA double-strand breaks (DSBs) arise programmatically during meiosis, V(D)J recombination, and immunoglobulin class switching and serve important functions therein[1–4]. DSBs are also induced by reactive metabolites, myriad chemotherapeutic agents, and upon exposure of cells to radiation and mutagenic chemicals[5,6]. The persistence or mis-repair of DSBs can result in cell death or chromosomal rearrangements characteristic of cancer cells[7].

DSBs are eliminated mainly via nonhomologous end joining (NHEJ) and homologous recombination (HR). In NHEJ, the break ends are processed and ligated, whereas homologous recombination (HR) entails the engagement of the sister chromatid to template repair[8–10]. NHEJ is the predominant pathway in the G1 phase of the cell cycle, whereas HR becomes fully functional in the S and G2 phases when the sister chromatid becomes available. The decision to repair a DSB by NHEJ or HR is made at the onset of DNA end resection, a process in which nucleases degrade the 5′-terminated strands of the DSB, and that is influenced by a variety of factors including the cell cycle stage and lesion context[11]. The 3′ ssDNA tails stemming from resection serve as the template for assembling the HR machinery harboring the recombinase RAD51, which searches for and invades the homologous region in the sister chromatid to form a displacement loop (D-loop), followed by DNA synthesis, resolution of recombination intermediates, and ligation to complete repair[8,9]. Occasionally, the resected DNA ends can also undergo alternative end joining (alt-EJ), in which break repair is guided by annealing of short homologous regions in separate 3′ DNA tails[12,13]. As such, alt-EJ invariably generates deletions and is therefore highly mutagenic, and this often characterizes DNA breakpoints in cancer-associated chromosome translocations[14]. Thus, the choice of DSB repair pathway can have a strong impact on the maintenance of genome integrity.

Several conserved nucleases participate in the DNA end resection process. The MRE11-RAD50-NBS1 (MRN) complex, containing the MRE11 nuclease, works with CtIP to initiate end resection by introducing a nick in the 5′ strand adjacent to DNA ends occluded by the NHEJ factor Ku[15–18]. MRN-CtIP can, subsequently, create a DNA gap via the 3′ to 5′ exonuclease activity of MRE11[19,20]. Long-range 5′ strand resection initiated from the DNA nick or gap generated by MRN-CtIP is catalyzed either by the 5′ to 3′ exonuclease EXO1 or the endonuclease DNA2 in conjunction with the BLM helicase[21–24]. In the latter case, BLM separates DNA strands to generate a 5′ DNA flap for cleavage by DNA2[25–27].

The selection pressure that has driven eukaryotes to maintain seemingly redundant resection pathways has remained an enigma. In this study, we begin to address this issue by investigating the nucleolytic processing of DSB ends with associated non-canonical nucleotides. The most common aberrant bases are ribonucleotides inserted by DNA polymerases or that stem from the deamination of cytosine residues to uracil[28]. RNA segments of 8–12 nt can stem from incomplete removal of RNA primers during lagging strand DNA synthesis[29–31]. R-loop structures, which frequently arise during transcription, harbor RNA:DNA hybrids ranging in size from less than 100 bp up to 2 kb[32,33]. Moreover, de novo RNA polymerase II-mediated transcription has been shown to create RNA:DNA hybrids at DSB sites[34–36]. Reactive oxygen species generated during cellular metabolism induce lesions such as 8-oxo-guanine[37]. Furthermore, the processing of uracil, 8-oxo-G, and other damaged bases leads to the formation of apurinic/apyrimidinic (AP) sites[38]. As such, the DNA end resection machinery is expected to encounter ribonucleotides and different DNA lesions. In this study, we show that lesion context differentially influences the action and efficiency of the long-range resection machineries. Biological experiments

yield evidence for a capability of the BLM-DNA2 ensemble to negotiate past these DNA lesions, and they also attest to the pivotal role of RNase H2 in the clearance of RNA–DNA hybrids during resection. These findings provide insights into pathway crosstalk in DSB repair and help explain why eukaryotes engage multiple nucleases in DNA end resection.

## Results

**Stimulation of EXO1 by RNA.** RNA:DNA hybrids can arise at DSB sites under different circumstances. Collision of a replication fork with a DNA lesion may lead to the formation of a DSB with 5′ terminal RNA that is a remnant of the RNA primer from lagging strand DNA synthesis. Additionally, recent studies have shown that DSB formation is accompanied by de novo transcription[39–41], which is expected to give rise to a varying length of 5′ terminated RNA at the break termini. Moreover, the collapse of a replication fork stalled at an R-loop would result in a DSB with an adjacent RNA:DNA hybrid[32,33]. To ask how such RNA content might affect end resection, we tested human EXO1 on 3′ labeled 30-mer dsDNA substrates containing 1, 4, or 10 nt of RNA at the 5′ end. Surprisingly, 1 or 4 nt of 5′ RNA led to an increase in EXO1 activity (Fig. 1a, lanes 1-15). EXO1 was able to resect through 10 nt of RNA with about the same efficiency as the control dsDNA substrate (Fig. 1a, lanes 16–20). These data suggest that short stretches of 5′ RNA stimulate EXO1. Importantly, we observed the same stimulatory effect of RNA placed at the 5′ terminus with yeast Exo1 (Supplementary Fig. 1a). To prevent degradation of the complementary unlabeled strand, we also performed the assay on substrates containing a 4-nt 5′ DNA overhang, which inhibits EXO1 activity. In this case, EXO1 was similarly stimulated by a 4-nt RNA segment placed at the 5′ terminus of the susceptible strand (Supplementary Fig. 1b).

The nuclease-deficient EXO1-D173A mutant[42], purified in exactly the same manner as the wild type counterpart, failed to cleave the substrate containing 4 nt of 5′ RNA (Fig. 1a, lanes 21–30), arguing against the possibility of an RNase contamination in our EXO1 preparations. A truncated version of human EXO1 containing the nuclease domain alone (residues 1–346), which was purified from S. cerevisiae, was also stimulated by 5′ RNA (Supplementary Fig. 1c, lanes 1–10), as was a similar fragment (residues 1–352) purified from E. coli (Supplementary Fig. 1c, lanes 11–20; gift from Lorena Beese). λ Exonuclease, which shares the 5′ to 3′ polarity of EXO1 but is structurally distinct, was blocked by RNA placed at either the 5′ terminus or at an internal location (Supplementary Fig. 1d). Together, these data show that the ribonuclease activity we observed is intrinsic to EXO1 and that it is conserved in the yeast ortholog.

DSB ends are rapidly occluded by the DNA end binding factor Ku in cells. Herein, resection is initiated when the MRN-CtIP complex creates a nick adjacent to the Ku-bound DSB end. This nick serves as the entry site for EXO1 or BLM-DNA2[19]. We therefore asked whether stimulation of EXO1 by 5′ RNA also occurs at a nick. Again, strong stimulation of resection occurred when a 4-nt RNA segment was situated at the nick site (Fig. 1b). We also tested the effect of RNA:DNA hybrid at DSBs containing a 3′ overhang, likely to be present at DSBs at a broken DNA replication fork. We note that a DNA substrate with a 5-nt 3′ overhang was acted on with higher efficiency by EXO1 than an equivalent substrate without such an overhang (Fig. 1a, lanes 1–5, vs. Fig. 1c, lanes 1–5). Importantly, the addition of 1 or 4 nt of 5′ RNA at the recessed end stimulated EXO1 activity (Fig. 1c, lanes 6–15). Unlike at a blunt end, however, stimulation was also observed upon extending the RNA to 10 nt (Fig. 1c, lanes 16–20). Given that lagging strand RNA primers are typically 8–12 nt long[43], these data suggest that EXO1 is more adept at resecting

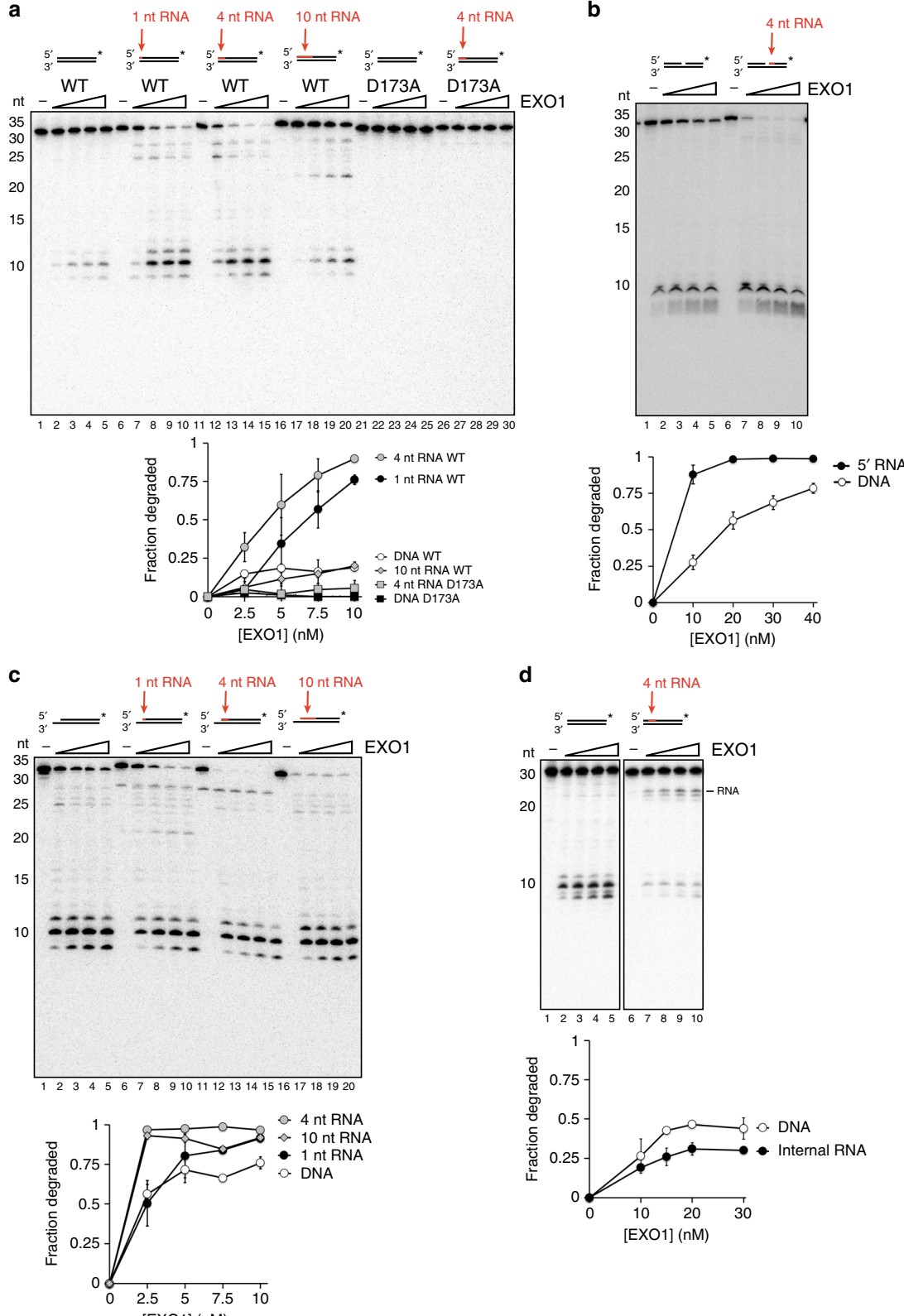

**Fig. 1 Effect of ribonucleotides on EXO1 activity. a** EXO1 (2.5, 5, 7.5 10 nM) or the nuclease-deficient EXO1-D173A mutant was incubated with the indicated 30-mer substrate (2.5 nM) at 37 °C for 10 min. Products were resolved in a 20% denaturing polyacrylamide gel. **b** Nicked substrates with or without a 4-nt RNA tract were tested as in (**a**). These substrates consisted of two 35-mers annealed to a 70-mer to create the central nick. **c** The indicated substrates with a 5-nt 3′ overhang were tested as in (**a**). **d** The indicated substrates with a 4-nt RNA tract located 4 nt internally from the 5′ end were tested as in (**a**). Error bars in all the panels represent the standard deviation of results from $n = 3$ independent experiments, and all points represent the mean. RNA is denoted in red, and the asterisk indicates the location of the radiolabel.

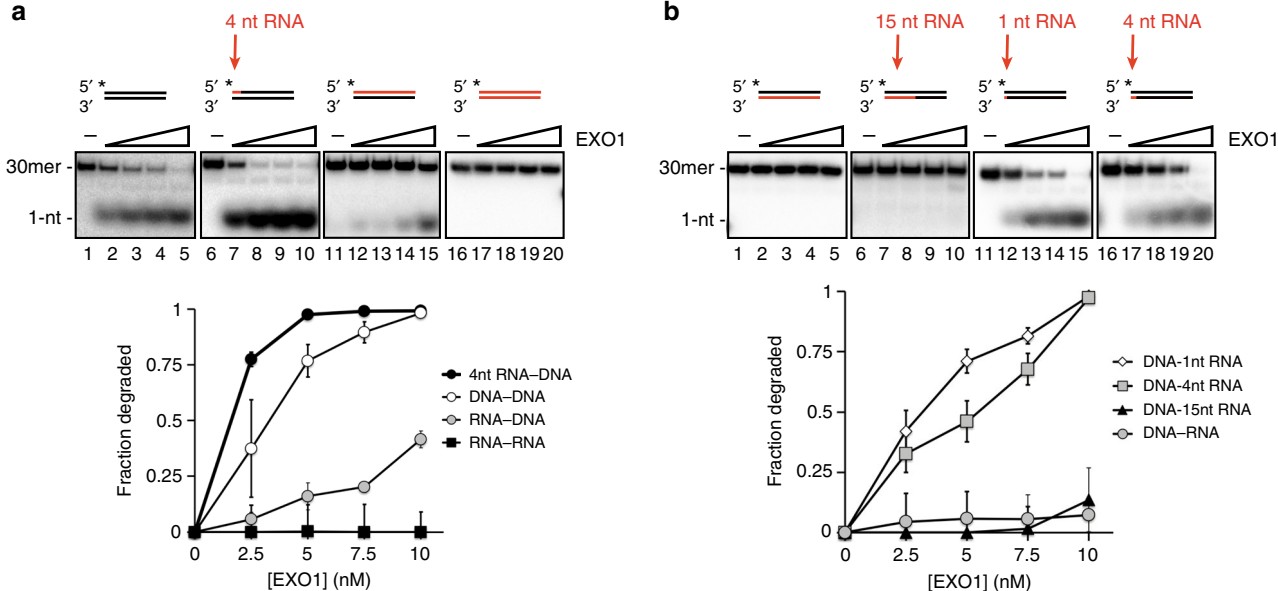

**Fig. 2 Effect of ribonucleotides on EXO1 initiation.** a The indicated 5′ end-labeled substrates were incubated with EXO1 (2.5, 5, 7.5, 10 nM) and reaction mixtures were resolved in a 10% native acrylamide gel. **b** The indicated 5′ end-labeled substrates were tested as in (**a**). Error bars in all the panels represent the standard deviation of results from $n = 3$ independent experiments, and all points represent the mean. RNA is denoted in red, and the asterisk indicates the location of the radiolabel.

replication-associated DSB ends containing unprocessed remnants of Okazaki fragments.

To further investigate how the position of the RNA affects EXO1, we created a substrate in which an internal run of 4 ribonucleotides was placed 4 nt from the 5′ end (Fig. 1d). Surprisingly, this internal RNA inhibited EXO1 and led to formation of a cleavage product corresponding to the location of the RNA (Fig. 1d). However, the RNA segment did not completely block EXO1, as smaller products were still observed (Fig. 1d).

We next radiolabeled the 5′ end of RNA-containing substrates to determine whether RNA stimulates resection initiation. The results revealed that digestion of the first 5′ nucleotide of a 4-nt RNA-containing substrate occurs more efficiently (Fig. 2a, lanes 1–10), but that extending the RNA to create a full RNA:DNA hybrid leads to inhibition (Fig. 2a, lanes 11–15). EXO1 showed no activity on dsRNA (Fig. 2a, lanes 16–20).

Interestingly, EXO1 failed to digest the 5′ DNA end of a full DNA-RNA hybrid (Fig. 2b, lanes 1–5). In fact, the presence of 15 nucleotides of RNA in the complementary strand led to near abrogation of DNA digestion (Fig. 2b, lanes 6–10). While a single ribonucleotide had no discernible effect (Fig. 2b, lanes 11–15), slight inhibition was observed when the RNA length was increased to four nucleotides (Fig. 2b, lanes 16–20). Together, these data show that even though the presence of a short 5′ RNA tract stimulates resection, RNA in the complementary strand restricts EXO1 activity on the 5′ DNA terminus.

**Bypass of RNA by BLM-DNA2.** We also examined BLM-DNA2 for its ability to resect DNA substrates that harbor ribonucleotides. First, we tested 3′ end-labeled substrates containing 4 nt of RNA at the 5′ terminus or at an internal location of the labeled strand. RNA in either location had little effect on resection efficiency by the BLM-DNA2-RPA ensemble, although a dramatic change in the cleavage pattern was observed when RNA was present internally (Fig. 3a, lanes 6–10). Whereas DNA2 generated multiple cleavage products in the middle of the dsDNA substrate (Fig. 3a, lanes 1–5), it incised primarily 1 nt after the embedded

RNA to produce a 15-nt product (Fig. 3a, lanes 6–10). This suggests that DNA2 is unable to cleave within RNA but can act on neighboring DNA.

We next sought to determine the effects of RNA on BLM and DNA2 by testing the two enzymes on RNA-containing substrates separately. We first investigated how an RNA:DNA hybrid affects the BLM helicase. A 4-nt tract of RNA at the 5′ end slightly stimulated BLM activity, whereas RNA placed at the internal locale attenuated unwinding (Fig. 3b). These inhibitory effects of RNA were modest, however, and overall activity was comparable to that on dsDNA (Fig. 3b). BLM showed substantially reduced activity on a full RNA:DNA hybrid and was inactive toward dsRNA (Fig. 3c). BLM translocates in the 3′ to 5′ direction while unwinding duplex DNA and shows the greatest activity on substrates with 3′ ssDNA overhangs[44]. We hypothesized that the reduction in BLM activity on the RNA:DNA hybrid (Fig. 3c) might reflect an inability of the helicase to translocate on the RNA strand. To test this, we constructed a substrate with a 5-nt 3′ overhang on one strand, which serves as an entry point for BLM to initiate translocation. Two versions were made in which the overhang-containing strand consisted of either RNA or DNA. While BLM efficiently unwound the DNA substrate, it was barely active on the RNA:DNA hybrid (Fig. 3d). Together, these data show that BLM can negotiate through a short RNA patch while unwinding DNA, but becomes strongly inhibited when the strand consists only of RNA, likely reflective of an inability of BLM to translocate on RNA. In agreement with this idea, the ATPase activity of BLM became strongly attenuated when the ssDNA cofactor was replaced with ssRNA of the same sequence (Supplementary Fig. 2a).

To test whether DNA2 is able to cleave within an embedded RNA tract, we constructed Y-shaped substrates that mimic the intermediate generated as a result of strand separation by BLM. These substrates contain a 31-nt dsDNA region with 44-nt overhangs of either DNA or with the 5′ DNA overhang harboring an internal patch of four ribonucleotides located 21 nt away from the duplex junction. Whereas DNA2 generated cleavage products throughout the 5′ DNA flap (Fig. 4a, lanes 1–7), it did not act on the embedded RNA as indicated by the lack of any product in the

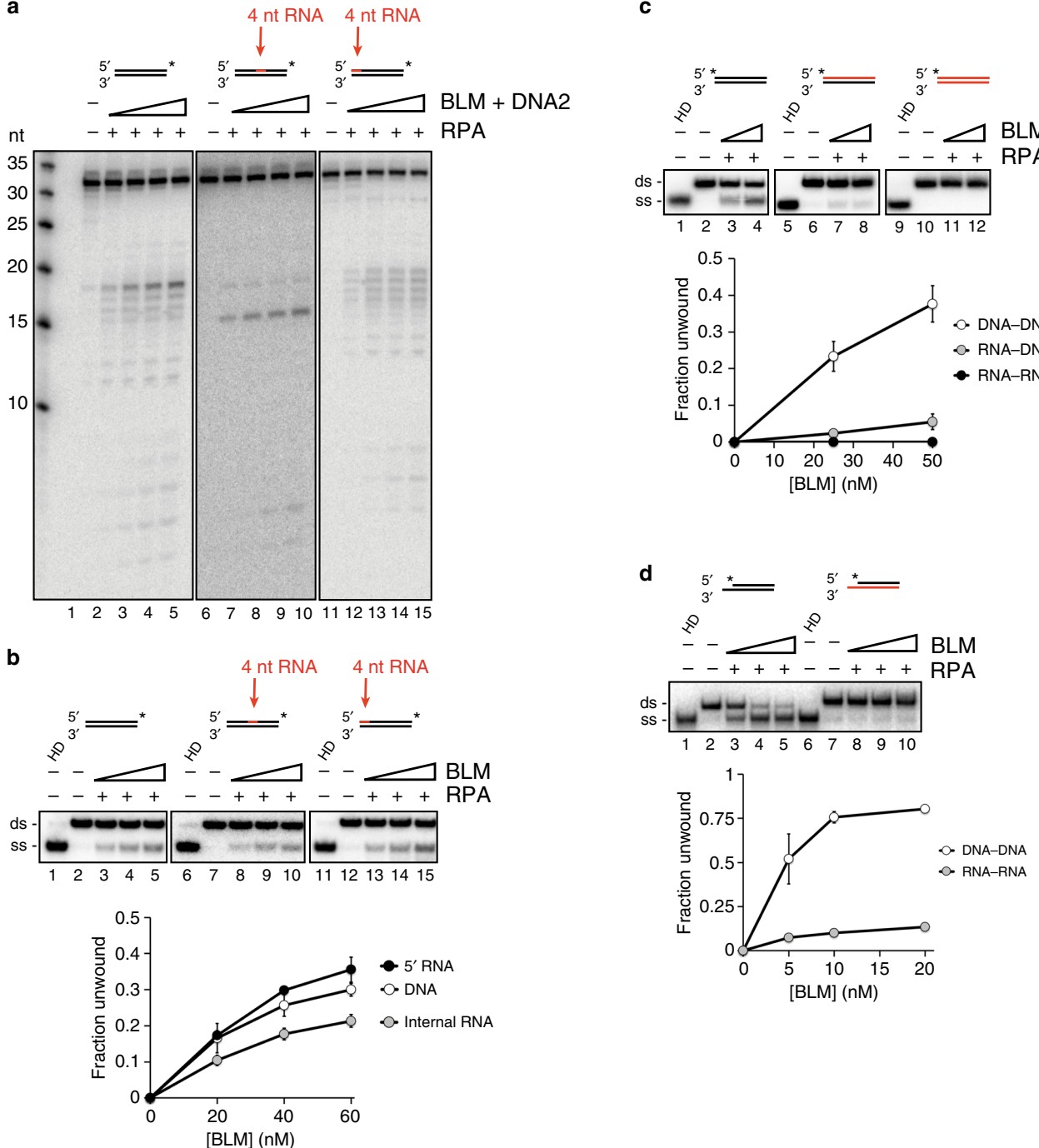

**Fig. 3 Effect of ribonucleotides on resection by BLM-DNA2. a** BLM and DNA2 (10, 20, 30, 40 nM) along with RPA (12.5 nM) were incubated with the indicated substrate (2.5 nM) at 37 °C for 10 min. Electrophoresis and quantification were performed as in Fig. 1a. **b** DNA unwinding by BLM (20, 40, 60 nM) was monitored in the presence of RPA (12.5 nM). Reaction mixtures were resolved in a 10% native acrylamide gel. In the lane labeled HD, the substrate was heat denatured by incubation at 95 °C for 2 min. The dsDNA and ssDNA bands were quantified and data were graphed. **c** Substrate unwinding by BLM (25, 50 nM) in the presence of RPA (12.5 nM) was examined as in (**b**). **d** Substrate unwinding by BLM (5, 10, 20 nM) was examined in the presence of RPA (12.5 nM) as in (**b**). Error bars in all the panels represent the standard deviation of results from $n = 3$ independent experiments, and all points represent the mean. RNA is denoted in red, and the asterisk indicates the location of the radiolabel.

52–56 nt range (Fig. 4a, lanes 8–14). Instead, a prominent product band of ~45 nt was seen (Fig. 4a, lanes 8–14), indicating that cleavage occurred in the DNA region downstream of the RNA. For clarity, an enlarged version of the relevant region of the gel from Fig. 4a is shown in Supplementary Fig. 2b. A similar result was obtained in the presence of RPA (Fig. 4b) and with yeast Dna2 (Supplementary Fig. 2c). These data are consistent

with our results from examining BLM-DNA2-RPA (Fig. 3a) and indicate that DNA2 cannot incise within RNA.

Previous studies have shown that RNA on the 5′ terminus of a flap stimulates yeast Dna2, but has no effect on the *Pyroccocus horikoshii* Dna2 ortholog[45,46]. We therefore tested whether a 4-nt segment of 5′ terminal RNA would affect flap cleavage by human DNA2. The human nuclease cleaved the RNA-containing

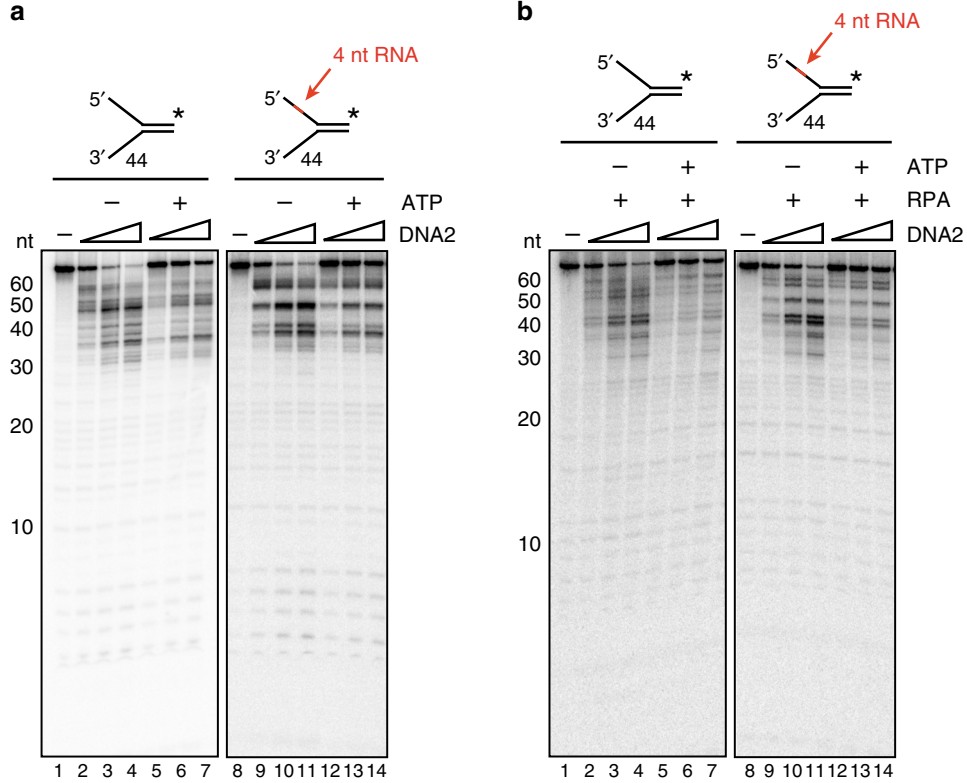

**Fig. 4 DNA2 cannot cleave within stretches of RNA. a** DNA2 (0.25, 0.5, 0.75 nM) was incubated with the indicated Y-shaped substrate (2.5 nM) at 37 °C for 5 min in the presence or absence of 2 mM ATP. Reaction mixtures were resolved in a 20% denaturing polyacrylamide gel. **b** The same assay was done in the presence of RPA (10 nM). Reaction mixtures were resolved in a 20% denaturing polyacrylamide gel. Error bars in all the panels represent the standard deviation of results from $n = 3$ independent experiments, and all points represent the mean. RNA is denoted in red, and the asterisk indicates the location of the radiolabel.

substrate with the same efficiency as the comparable DNA flap, regardless of the presence of RPA (Supplementary Fig. 2d). Thus, 5′ terminal RNA does not stimulate human DNA2.

**Differential effects of 8-oxo-G and AP site on EXO1 and DNA2**. We next asked whether EXO1 could resect through two common base lesions, namely, 8-oxo-guanine, the most abundant oxidized base in the human genome, and the AP site, created when a damaged base or uracil residue is removed by a lesion-specific DNA glycosylase or Uracil DNA Glycosylase, respectively. For this, 30-mer dsDNA substrates were created in which the lesion in question was situated at position 11 from the 5′ terminus. While the AP site had little or no effect on EXO1 (Fig. 5a, lanes 1–10), the 8-oxo-G residue caused the nuclease to stall about 4–5 nt upstream of the lesion (Fig. 5a, lanes 11–15). With higher EXO1 concentrations, we also observed a product corresponding to stalling at the lesion site (Fig. 5a, lanes 14–15). These data suggest that 8-oxo-G residues that occur near DSB sites impede EXO1-dependent resection.

Interestingly, the presence of an AP site strongly stimulated resection by BLM-DNA2, while significant enhancement by the 8-oxo-G lesion was also seen (Fig. 5b). We hypothesized that disruption of base pairing by the AP site might allow BLM to engage and unwind the substrate more readily. To test this idea, we constructed another substrate in which the AP site was replaced by adenine to create an A-C mismatch. Consistent with the above premise, the DNA mismatch also enhanced DNA unwinding by BLM, albeit to a lesser degree when compared to the AP lesion (Fig. 5c, lanes 1–10, 16–20). In contrast, 8-oxo-G, which can base pair with the opposing cytosine but increases

DNA bending[47], had little or no effect on BLM-mediated DNA unwinding (Fig. 5c, lanes 11–15). Thus, in DSB end resection, adjoining lesions that disrupt base pairing may facilitate DNA separation by BLM to create the requisite flap structure for DNA2 to act on.

**Resection through base lesions and RNA:DNA hybrids in cells.** To determine whether the differences in substrate specificity between EXO1 and BLM-DNA2 that we observed in vitro are relevant to resection in cells, we manipulated cellular conditions to increase the burden of 8-oxo-G residues, AP sites, or RNA:DNA hybrids by performing siRNA knockdown against 8-oxo-G DNA Glycosylase (OGG1)[48,49], Apurinic-Apyrimidinic Endonuclease 1 (APE1)[50,51], or RNase H2A[32,33,52], respectively.

We first irradiated U2OS cells following siRNA-mediated knockdown of EXO1 or DNA2 and assessed DNA end resection by quantifying RPA foci[53]. Knockdown of only EXO1 or DNA2 resulted in significant impairment of resection (Supplementary Fig. 3a), in concordance with the established role of these nucleases in long-range resection[21,22,54]. Next, we similarly tested cells with knockdown of OGG1 or APE1 along with additional depletion of either EXO1 or DNA2. Interestingly, radiation-induced RPA foci numbers were higher in cells with OGG1 depletion, which might be due to increased DNA breakage as OGG1 loss has been shown to result in accumulation of single-strand breaks[55]. Supplementary Fig. 3b; Fig. 6a). Importantly, additional depletion of EXO1 resulted in only a slight reduction in radiation-induced RPA foci, while DNA2 knockdown almost completely abrogated resection (Fig. 6a). Similarly, in cells depleted for APE1 (Supplementary Fig. 3c), additional

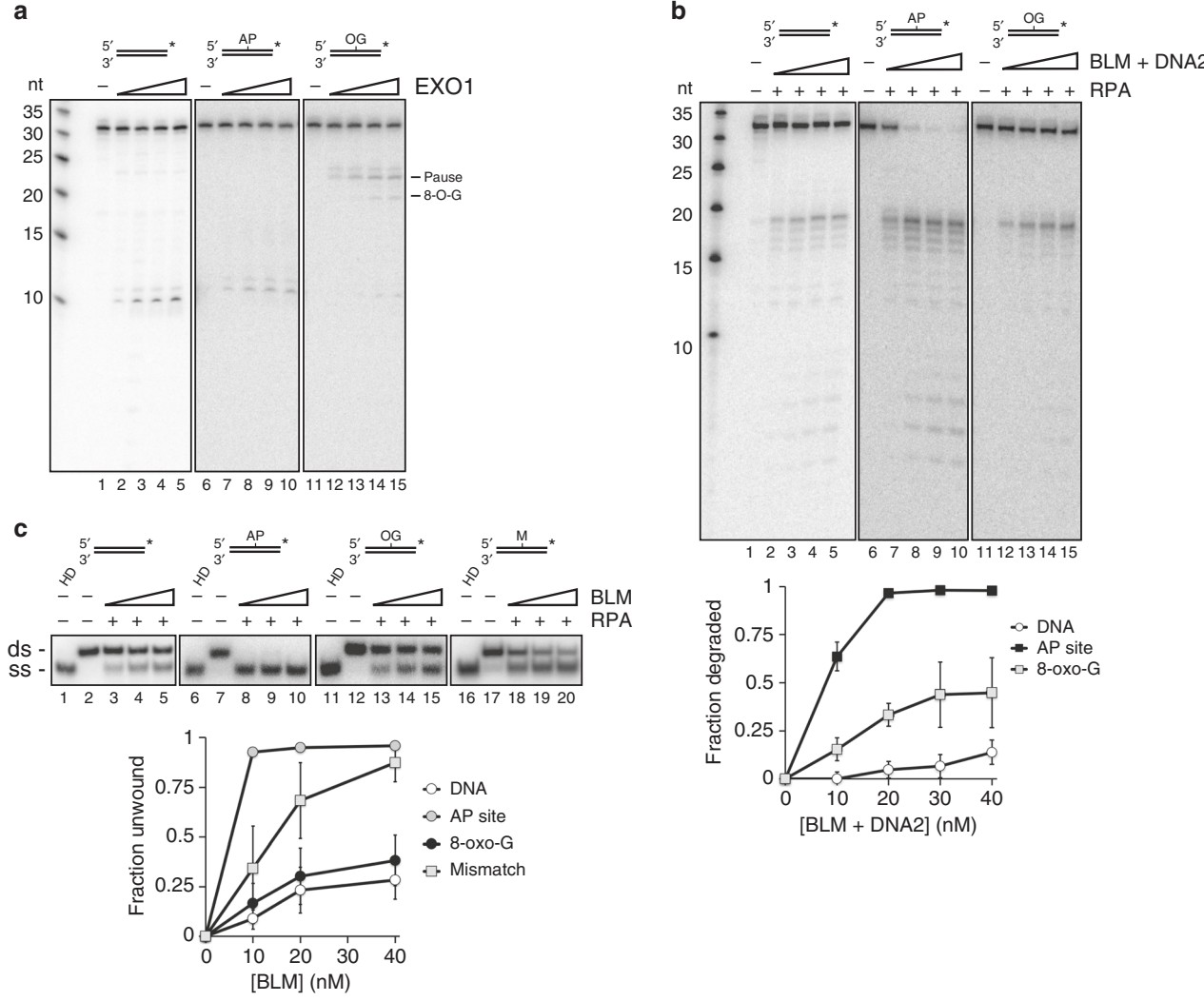

**Fig. 5 Effect of 8-oxo-G and AP site on resection by EXO1 and BLM-DNA2. a** EXO1 (10, 20, 30, 40 nM) was incubated with the indicated substrate (2.5 nM) at 37 °C for 10 min. The 8-oxo-G and AP site are indicated by OG and AP, respectively. Reaction mixtures were resolved in a 20% denaturing acrylamide gel. **b** BLM, DNA2 (10, 20, 30, 40 nM) and RPA (12.5 nM) were tested on the same substrates in (**a**). **c** DNA unwinding by BLM (20, 40, 60 nM) was examined in the presence of RPA (12.5 nM) as in Fig. 3b. Error bars in all the panels represent the standard deviation of results from $n = 3$ independent experiments, and all points represent the mean. In some cases, the error bars are smaller than the symbol and are thus not easily discernable. RNA is denoted in red, and the asterisk indicates the location of the radiolabel.

knockdown of EXO1 resulted in a small reduction in RPA foci, while knockdown of DNA2 led to a more significant impairment of focus formation (Fig. 6b). These data suggest that DNA2 plays a more prominent role in the resection of DNA with an increased burden of 8-oxo-G residues or AP sites. These findings are in congruence with our biochemical data showing that 8-oxo-G residues specifically impede EXO1 while upregulating resection efficiency by BLM-DNA2 (Figs. 5a, b), and that AP sites stimulate unwinding by the DNA2-associated helicase BLM (Fig. 5c).

Finally, we evaluated resection in cells with knockdown of RNase H2A, which is critical both for the removal of interspersed ribonucleotides in DNA as well as for the degradation of the RNA component of R-loop structures[32,33,52]. Interestingly, radiation-induced RPA foci were significantly reduced in cells with RNase H2A depletion, reflective of an overall impairment of resection (Supplementary Fig. 3d; Fig. 6c). Importantly, additional knockdown of EXO1 or DNA2 reduced RPA foci modestly. Taken together, our results suggest that neither EXO1 nor BLM-DNA2 is able to efficiently resect through RNA–DNA hybrids that are targeted by RNase H2A. We note that these data are in general

concordance with our biochemical data showing that EXO1 and BLM-DNA2 are impeded by 5′ terminated RNA tracts with greater than ten ribonucleotides or a short RNA tract present internally (Figs. 1d, 2a, 3c).

## Discussion

Multiple conserved DSB end resection pathways exist in eukaryotes from fungi to humans, but why such redundancy is necessary or desirable has remained unclear. It has been proposed that these resection pathways might be suited for processing DNA breaks that harbor different lesions or are differentially engaged in various cell cycle stages[54]. Here, we have documented properties of EXO1 and BLM-DNA2 at DSBs with physiologically relevant base lesions that help shed light on this issue. The principal findings of this study, summarized in Fig. 6d, are (1) a short RNA tract located near the 5′ terminus enhances EXO1 activity, but an internal tract of RNA or a long RNA:DNA hybrid inhibits EXO1; (2) BLM efficiently unwinds DNA containing a short stretch of RNA, but is impeded by a long RNA:DNA hybrid;

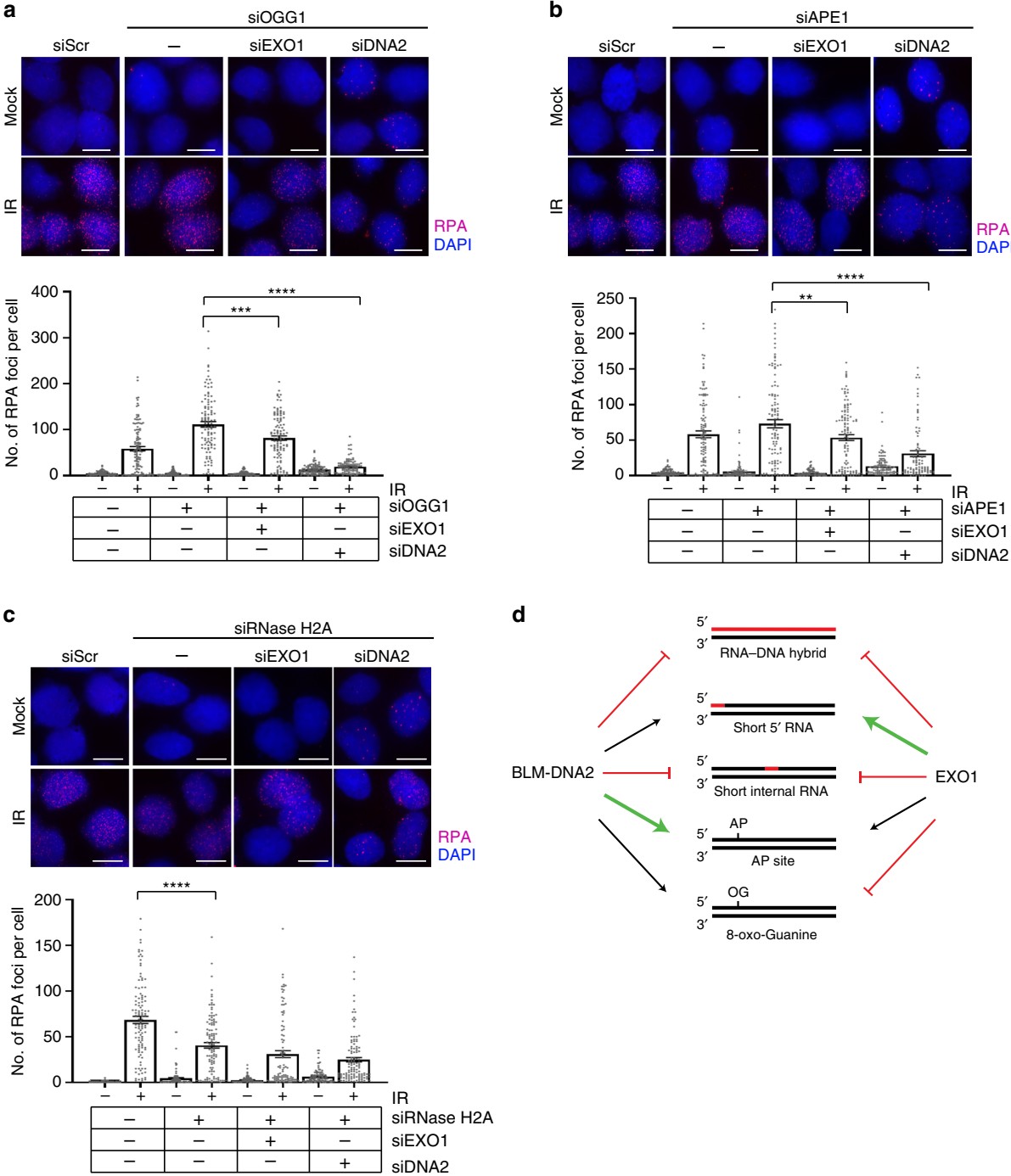

**Fig. 6 Biological effects of 8-oxo-G residues, AP sites, and ribonucleotides. a** U2OS cells were depleted of OGG1 in combination with EXO1 or DNA2 knockdown using siRNA as indicated. Cells were screened for resection defects by quantifying IR-induced RPA foci 3 h after treatment with 6 Gy IR. Representative images of mock-irradiated or irradiated (IR) cells immunostained with anti-RPA antibody (red) are shown. Nuclei were stained with 4′,6-diamidino-2-phenylindole (blue). Plot shows average numbers of RPA foci per nucleus. **b** U2OS cells were depleted of APE1 in combination with EXO1 or DNA2 knockdown using siRNA, and assayed for resection defects as described above. **c** U2OS cells were depleted of RNase H2A in combination with EXO1 or DNA2 knockdown using siRNA, and assayed for resection defects as described above. All experiments were replicated $n = 3$ times, and all bars represent the mean. Error bars depict standard error of the mean. ** indicates $p = 0.0068$; ***, $p = 0.0002$; ****, $p < 0.0001$; two-tailed unpaired-t-test. The scale bar represents 10 μm for all images. **d** Model summarizing the effect of ribonucleotides, AP sites, and 8-oxo-G residues on resection. Black arrows indicate activity similar to that on control substrates devoid of any lesion, bold green arrows indicate stimulation of resection, and red perpendicular lines indicate attenuation of activity.

(3) DNA2 cannot act on RNA located in a 5′ flap but can efficiently cleave an adjoining DNA region; (4) EXO1 can digest an AP site but is impeded by 8-oxo-G; (5) an AP site stimulates BLM activity; (6) conditions that increase the load of 8-oxo-G residues or AP sites in cells (i.e. loss of OGG1 or APE1) result in increased dependence on the DNA2-dependent resection pathway; and (7) loss of RNase H2A reduces resection efficiency in cells, likely due to the accumulation of RNA:DNA hybrids that are inhibitory to both EXO1 and BLM-DNA2 at DSB sites. Altogether, our biochemical and cell-based analyses provide evidence that the apparently redundant DNA end resection pathways are in fact adept for specific lesion contexts.

Importantly, a short RNA tract stimulates EXO1, but inhibition occurs when the RNA length is increased (Fig. 1). A long RNA:DNA hybrid tract causes the double helix to adopt the A form, which could explain the observed inhibitory effect on EXO1. We do not yet understand how the presence of a short 5′ RNA tract upregulates EXO1 activity. One possibility is that contacts between the enzyme and the downstream region of the substrate may confer such specificity[42]. The observation that truncated EXO1 variants lacking the C-terminal domain are still stimulated by RNA (Supplementary Fig. 1c) suggests that protein–nucleic acid contacts crucial for this stimulatory effect are retained in the N-terminal nuclease domain. While a short 5′ terminal RNA segment stimulates EXO1 (Fig. 1a–c), such an internal RNA tract causes EXO1 to pause (Fig. 1d). The mechanistic basis of the inhibitory effect of internally situated RNA tracts remains to be delineated.

The observation that 3′ RNA blocks EXO1 activity on the abutting 5′ DNA end (Fig. 2b) hints at a mechanism for regulation of resection. Delaying resection might be desirable at DSBs undergoing ongoing transcription to maintain the integrity of the DNA template until transcription is complete. Similarly, DSBs generated by R-loop cleavage might be left unresected until an alternate mechanism of resolution is engaged. We note that neither EXO1 nor BLM-DNA2 is adept at resecting long RNA:DNA hybrids, which might be found at R-loop associated DSBs. In this case, the recruitment of an RNA:DNA helicase, such as SETX[56], may lead to the clearance of such RNA:DNA hybrids.

We have demonstrated that, unlike EXO1, the BLM-DNA2 pathway bypasses regions of RNA to act on DNA downstream. This is consistent with a previous report showing that yeast Dna2 can bypass a 12 nt 5′ RNA primer when cleaving Okazaki fragments[45]. The BLM helicase efficiently unwinds substrates with a short RNA:DNA hybrid (Fig. 3b) but is inactive on dsRNA (Fig. 3c). That BLM is unable to unwind a substrate with a 3′ RNA overhang (Fig. 3d) suggests that BLM has minimal ability to translocate on RNA.

We have shown biochemically that AP sites and 8-oxo-G residues, being among the most common base lesions in the genome[57,58], exert differential effects on the two long-range resection pathways. While EXO1 is able to traverse through an AP site, it is strongly impeded by 8-oxo-G (Fig. 5a). Interestingly, the AP site strongly stimulates end resection by BLM-DNA2 and DNA unwinding by BLM, while 8-oxo-G also exerts a stimulatory effect on resection, albeit to a lesser degree compared to that seen with the AP site (Fig. 5b, c). Previous studies have shown that the AP site and 8-oxo-G lower the thermal stability of DNA[59,60]. Our observation that a DNA mismatch also stimulates unwinding by BLM (Fig. 5c) suggests that this stimulation is likely mediated via transient unpairing of dsDNA to create a 3′ flap from which BLM can initiate unwinding. Thus, our biochemical reconstitution studies suggest that duplex-destabilizing base lesions within the vicinity of a DSB are likely to promote helicase-mediated resection.

Predictions stemming from our biochemical analyses have been tested in the physiological setting by evaluation of DNA end resection in irradiated U2OS cells with siRNA-mediated knockdown of OGG1, APE1, or RNase H2, in conjunction with knockdown of either EXO1 or DNA2. As expected from past studies in yeast and human cells[21,22,54,61–63], we have verified that resection of ionizing radiation (IR)-induced breaks is equally dependent upon EXO1 and DNA2 in U2OS cells (Supplementary Fig. 3a). However, upon knockdown of OGG1 in these cells, resection becomes disproportionately dependent on DNA2 (Fig. 6a). This is consistent with our biochemical finding that 8-oxo-G specifically impedes EXO1 (Fig. 5a) while exerting a stimulatory effect on the BLM-DNA2 resection pathway (Fig. 5b). Likewise, following the knockdown of APE1, which leads to the accumulation of AP sites in DNA, we have observed a greater dependency on DNA2 relative to EXO1 for resection of IR-induced breaks (Fig. 6b). In congruence with these data, we have found stimulation of BLM-DNA2 by AP sites in vitro (Fig. 5b, c). These results provide biological validation that the HR-specific DNA resection machinery is endowed with the wherewithal of removing certain DNA base lesions.

RNase H2A plays an important role in the clearance of RNA:DNA hybrids, such as those associated with R-loops[64,65] or stemming from de novo transcription induced at DSB sites[40,66]. It also acts on ribonucleotides embedded in DNA to initiate ribonucleotide excision repair[67]. Loss of RNase H2A is therefore expected to increase the RNA content in DNA. We found that extended RNA:DNA regions are refractory to resection by both EXO1 and BLM-DNA2 in vitro (Figs. 2a and 3c), whereas individual and short stretches of ribonucleotides either stimulate EXO1 (if located at the 5′ end) (Fig. 1a–c) or lead to pausing if located internally (Fig. 1d). Internal ribonucleotides can be efficiently resected by the BLM-DNA2 pathway, however, with DNA2 cleaving downstream within the adjacent DNA (Fig. 4). Depletion of RNase H2A significantly impairs the resection of IR-induced breaks, and knockdown of either EXO1 or DNA2 has a modest effect on resection in RNase H2A-depleted cells (Fig. 6c). We speculate that the impaired resection upon depletion of RNase H2A is primarily caused by the accumulation of R-loops or RNA:DNA hybrids stemming from de novo transcription at DSBs[64,65]. It remains to be determined whether the involvement of RNase H2A in ribonucleotide excision repair[67,68] is also germane for the regulation of DNA end resection.

Our findings are also relevant for studies on the repair of complex DSBs induced by accelerated ions of high linear-energy transfer (LET). Such ions are being increasingly used for cancer therapy due to better dose distribution and greater DNA damage inflicted relative to low-LET X-rays[69]. Moreover, as components of galactic cosmic radiation, these ions also carry an increased cancer risk relative to terrestrial radiation[70,71]. High-LET ions induce DSBs that are flanked by other types of base damage including oxidized bases and AP sites. These complex DSBs are poorly repaired by NHEJ and are more reliant on the HR pathway for their resolution, which occurs only inefficiently[72–74]. A better understanding of how long-range nucleases resect complex DSBs could lead to approaches to augment carbon ion therapy as well strategies to mitigate cancer risks from high-LET radiation.

Altogether, our studies provide evidence for a division of labor among resection nucleases that enables cells to efficiently process diverse lesions that arise endogenously and as a result of exposure to genotoxic agents. Moreover, they furnish the conceptual and experimental framework to define the relationship between the HR machinery and other DNA repair pathways as well as R-loop avoidance and the nucleolytic processing of complex DNA breaks.

## Methods

**Protein purification.** Affinity epitope-tagged forms of EXO1, yExo1, DNA2, yDna2, BLM, and RPA were expressed and purified using our established protocols[24,26,27]. DNA that codes for the EXO1(1–346) fragment was introduced into pESC-URA by gap repair and the protein was expressed and purified using the same procedure as the full length protein (846 amino acid residues). EXO1 (1–352) was a gift from Lorena Beese and was purified from *E. coli*[42].

**DNA substrates.** The substrates were made by mixing a radiolabeled oligonucleotide with a 2-fold excess of an unlabeled complementary oligonucleotide in TE buffer (20 mM Tris-HCl, 1 mM EDTA, pH 8.0) containing 50 mM NaCl and slow cooling from 95 °C to room temperature over the course of 3 h. Substrates were separated from unannealed oligonucleotides in a native 10% acrylamide gel in TBE buffer (0.1 M Tris, 90 mM boric acid, 1 mM EDTA, pH 8.0) at 4 °C, electroeluted from gel slices, and concentrated in an Amicon Ultracel 30K concentrator. DNA concentration was determined by UV absorbance. Oligonucleotide sequences are provided in Supplementary Table 1.

**Helicase and nuclease assays.** Helicase and nuclease assays were performed in buffer R (20 mM Na-HEPES, pH 7.5, 2 mM ATP, 0.1 mM DTT, 100 μg/ml BSA, 0.05% Triton-X 100, 2 mM MgCl$_2$, and 100 mM KCl) and contained 2.5 nM DNA molecules. Reactions were conducted at 30 °C for the times indicated in the figure legends. After the addition of SDS to 0.2%, proteinase K to 0.25 μg/μl, and 0.08% Orange G dye with a final glycerol concentration of 10%, the reaction mixtures were deproteinized for 5 min at 37 °C. Products were separated on acrylamide gels in TBE buffer as indicated in the figure legends. For the analysis of endonucleolytic cleavage products generated by DNA2, reaction mixtures were processed in the same manner but were fractionated under denaturing conditions with 7 M urea. Gels were dried onto Hybond membrane on top of Whatman filter paper (GE) and then analyzed in a BioRad Personal Molecular Imager FX phosphorimager. Quantitation was by measuring loss of the substrate signal.

**ATPase assay.** ATPase assay was performed at 37 °C[24]. Briefly, BLM was incubated with an ssDNA or ssRNA 35-mer of identical sequence (100 nM) in 10 μl of buffer [20 mM HEPES-KOH at pH 7.5, 5 mM Mg(C$_2$H$_3$O$_2$)$_2$, 1 mM DTT, 0.2 mM ATP, 0.4 μCi γ-$^{32}$P-ATP] for the indicated time. The reaction was terminated by adding 2 μL of the reaction to an equal volume of 0.5 M EDTA. Next, 1 μL of the reaction mixture was spotted onto a PEI cellulose TLC plate (Select Scientific), which was developed in with 0.15 M formic acid/0.15 M LiCl. The TLC plates were air-dried and then subjected to phosphorimaging analysis.

**Cell culture and irradiation of cells.** U2OS cells (obtained from the American Type Culture Collection) were maintained in α-MEM medium, supplemented with 10% fetal bovine serum and penicillin/streptomycin in a humidified atmosphere with 5% CO$_2$. Cells were irradiated using the CellRad X-ray irradiator (Precision X-ray).

**Transfection of cells.** Depletion of OGG1, APE1, RNase H2A, EXO1 or DNA2 was carried out by transfection with appropriate siRNAs (Supplementary Table 2) using Lipofectamine RNAiMAX (Invitrogen). Cells were harvested 72 h later to verify knockdown by western blotting. Cells were transfected with scrambled siRNA as control for all experiments.

**Western blotting and antibodies.** Nuclear extracts for Western blotting were prepared by resuspending cell pellets in hypotonic lysis buffer (10 mM Tris-HCl pH 7.5, 1.5 mM MgCl$_2$, 5 mM KCl, protease and phosphatase inhibitors), followed by nuclear extraction buffer (50 mM Tris-HCl pH 7.5, 0.5 M NaCl, 2 mM EDTA, 10% sucrose, 10% glycerol, protease and phosphatase inhibitors). Whole cell extracts were prepared by resuspending cell pellets in 1xSDS protein lysis buffer (67.5 mM Tris pH 6.8, 25 mM NaCl, 0.5 mM EDTA, 12.5% Glycerol, 0.25% SDS, DTT). The following primary antibodies were used: OGG1 (Abcam), APE1 (Santa Cruz Biotechnology), EXO1, RNase H2A (Bethyl), DNA2 (Thermo Fisher Scientific) and beta-actin (Cell Signaling). The following secondary antibodies were used: horseradish peroxidase-conjugated secondary antibodies (Bio-Rad) and Alexa488/568-conjugated secondary antibodies (Invitrogen). Antibody dilutions and catalogue numbers are provided in Supplementary Table 3. Uncropped gels are shown in Supplementary Figs. 4, 5.

**Immunofluorescence staining.** Cells were seeded onto glass chamber slides (Falcon) and immunostained with anti-RPA antibody 3 h after irradiation with 6 Gy. Cells were pre-extracted with buffer I (10 mM PIPES, pH 7.0, 100 mM NaCl, 300 mM Sucrose, 3 mM MgCl$_2$, 1 mM EGTA, 0.5% Triton X-100), extraction buffer II (10 mM Tris pH 7.5, 10 mM NaCl, 3 mM MgCl$_2$, 1% Tween-40 and 0.5% sodium deoxycholate). Cells were fixed with 4% paraformaldehyde/PBS and permeabilized with 0.5% Triton X-100 before incubation with antibodies. Images were captured using a Nikon Swept Field fluorescence microscope (×40 objective lens). The average number of RPA foci per nucleus was determined after scoring at least 100 nuclei. Images were generated in the Core Optical Imaging Facility which is

supported by UTHSCSA, NIH-NCI P30 CA54174 (CTRC at UTHSCSA). Statistical analysis was carried out using GraphPad Prism software.

**Reporting summary.** Further information on research design is available in the Nature Research Reporting Summary linked to this article.

## Data availability
All data generated or analyzed during this study are included in this published article (and its supplementary information files) and are available from the authors upon reasonable request

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

## Acknowledgements

This study was supported by NIH grants R35 CA241801, R21 ES027154, RO1 CA197796, RO1 ES007061, RO1 CA205224, RO1 GM109645, PO1 CA092584, and P30 CA054174, by the National Aeronautics and Space Administration Award NNX16AD78G, a Team Science Grant from the Gray Foundation under the Basser Initiative, and by CPRIT REI Award RR180029. PS is the holder of the Robert A. Welch Distinguished Chair in Chemistry (AQ-0012). S.B. is the holder of the Mays Family Foundation Distinguished Chair in Oncology.

## Author contributions

J.D., N.T., S.B., R.H., and P.S. conceived and designed experiments. J.D., N.T., and G.H. performed experiments. J.D., W.W., A.M., X.X., H.K., E.W., and K.N. generated expression constructs and purified proteins for the study. J.D., R.H., B.M., S.B., and P.S. analyzed the data and wrote the paper.

## Competing interests

The authors declare no competing interests.
