## [Peer Review File · Nature Communications]

Reviewers' comments:

Reviewer #1 (Remarks to the Author):

This is a very neat paper that demonstrates that long-range DNA end resection pathways respond differentially to non-canonical structures at DNA ends. This may explain the need for multiple resection pathways.

The key observations made are:

- EXO1 is stimulated by short RNA segments in the 5'-terminated strands, while inhibited by RNA in the 3'-terminated strand. This would make EXO1 suitable to degrade DNA ends resulting from broken replication forks where RNA segments are likely to be present

- BLM-DNA2, instead, is not significantly affected by RNA, but it appears promoted by AP sites in the template, which the authors explain by the duplex destabilization effect by this lesion

Generally, the experiments are performed qualitatively at a very high standard, as typical in this lab. My main concern is related to physiological relevance. The authors should attempt to individually inhibit RNaseH1-2, AP endonuclease or treat the cells with oxidative agents, and then monitor the relative contribution of the BLM-DNA2 and EXO1 pathways to resection; e.g., scoring for total RPA signal by FACS may be the best.

Other comments

- Is the preference for AP sites by BLM-DNA2 also apparent under reaction conditions when $[Mg] > [ATP]$, as the duplex destabilization might be artificially enhanced by the used conditions where there is no free magnesium

- Yeast Dna2 was reported to be stimulated by RNA at the 5' end by the Seo lab (Bae JBC, 2000), but this was disputed later using Pyrococcus Dna2 (Higashibata JBC, 2003). What would be the effect of terminal RNA in experiments such as in Fig. 5 with human DNA2?

Reviewer #2 (Remarks to the Author):

Various genetic studies in yeast and human cells revealed that BLM-DNA2 (Sgs1-Dna2) and EXO1 (Exo1) function in distinct but complementary pathways of DNA resection to promote homology-directed repair of DNA double-strand breaks (DSBs). Gravel et al. acknowledged that it may simply "provide cells with a buffer against genotoxic challenges" but also "speculated that the two pathways will turn out to be more complementary than redundant, with each pathway being best suited for different physiological conditions (e.g. cell cycle stage, different types of DSB lesions including clean DNA ends and DSBs containing complex base damage and/or difficult-to-process structures" (Gravel et al., 2008). But the question WHY cells possess two non-overlapping mechanisms of long-range resection remained largely unanswered?

In a serious attempt to tackle this question, Daley, Sung and co-workers describe a thorough in vitro analysis of the specificity of human EXO1 and BLM2-DNA2 resection machineries towards chimeric substrates containing either ribonucleotides (rNTs) or damaged bases, such as 8-oxoguanine (OG) or apurinic/apyrimidinic (AP) sites. In summary, they provide evidence that EXO1 nucleolytic processing is stimulated by the presence of short stretches of rNTs but specifically stalled upstream of a GO residue. In contrast, BLM-DNA2-mediated cleavage is prohibited by internal rNTs but untroubled or even enhanced by GO or AP sites, respectively. Based on these findings, the authors conclude that eukaryotic cells may have evolved two different

long-range resection machineries to compensate each other for the efficient nucleolytic digestion of non-canonical DNA structures present in the 5' strand.

Overall, I found the biochemical data and analysis to be clean and convincing and the manuscript to be well-written. Nonetheless, as outlined below, serious concerns remain as to whether the observed activities indeed reflect bona fide physiological functions of EXO1 and BLM2-DNA2 during DSB resection or enzyme-specific features irrelevant for DSB resection. Therefore, I consider this manuscript more suitable for a more technical journal (e.g. JBC)

Major comments:

- Conceptually, I find it a major flaw that the authors seem to neglect that cells are equipped with highly efficient and specialized DNA repair systems detecting and removing rNTs (RER, ribonucleotide excision repair) as well as GO and AP sites (BER, base excision repair) from anywhere in the DNA. Thus, it is equally possible that BER and RER cooperate with the HR pathway to remove these obstacles prior to DSB resection (or at least act in redundant fashion with either of the two long-range resection machineries). Along the same lines, EXO1 was shown to substitute for FEN1 during RER with reasonable efficiency (Sparks et al., 2012).
- The demonstration of an intrinsic 5'-3' RNase H-like activity of EXO1 is per se not original (Qiu et al., 1999, not cited here), and most likely reflects its physiological function during Okazaki fragment processing (again as a back-up for Rad27/FEN1; Kahli et al., 2019). In my opinion, none of the DNA/RNA substrates and assays used here are convenient to distinguish between in vitro 'resection of replication-associated DSB ends' or 'Okazaki fragment processing' by EXO1. The same applies to assays involving BLM-DNA2 ensemble. Therefore, I find the title highly speculative.

Additional comments:

- Key experimental data showing EXO1-dependent processing of substrates containing short internal RNA is missing (but included in the Summary, Fig 6d).
- It is unclear to me why BLM-DNA2-RPA-mediated substrate cleavage (Fig. 6b), which partially coincides with increased BLM-RPA-mediated substrate unwinding (Fig. 6c), is specifically stimulated by the presence of AP sites and GO. In fact, as hinted by the authors, the 'stimulation' is most likely due to alterations in the thermodynamic properties of the DNA substrates. Studies have shown that incorporation of AP sites and GO into DNA duplexes dramatically lower their thermal stability (e.g. Vesnaver et al., 1989; Singh et al., 2011), thereby potentially facilitating BLM-mediated unwinding and/or adjacent DNA2-mediated cleavage reactions carried out in vitro.
- Detailed information regarding the DNA substrates (e.g. synthesis, length, sequence) is completely missing.
- In Figure 2b, if this represents a denaturing gel, the length of unprocessed substrates cannot be 30 nt (as indicated by the marker on the left)?
- To properly measure 5'-3' exonuclease activities in EXO1 using short DNA duplexes, the 5' end of the unlabeled strand should be blocked (e.g. by adding a couple nucleotides, as EXO1 is highly unreactive towards ssDNA).
- Figure 1 is negligible

Reviewer #3 (Remarks to the Author):

In this interesting paper, the authors analyze biochemically the effect of two common DNA lesions, rNTPs embedded in a dsDNA and 8-oxo-G, in the processivity of the long-range resection nucleases EXO1 and DNA2/BLM. The authors show that, indeed, EXO1 is stimulated by rNTPs but blocked by 8-oxo-G, whereas DNA2/BLM shows the opposite preference. These are interesting results, that might explain why eukaryotic cells have maintained a plethora of different nucleases involved in DNA end resection and shed some light on how resection takes place in non-canonical dsDNA. However, my major concern is how relevant those observations are on an in vivo situation. It can be argued that the fact that an enzyme is stimulated in vitro by a specific DNA structure

might have no real biological meaning. Thus, in order to accept this work in Nat communication, I think some in vivo observations are required.

Thus, I would suggest the authors to try some in vivo resection assays in the biological system of their choice (as it seems the effect is conserved from yeast to humans) in which they test the relevance of EXO1 or DNA2/BLM in situations in which rNTPs, R-loops and 8-Oxo-G are artificially increased. More specifically, I suggest to analyze the effect of Exo1 and BLM/DNA2 depletion/KO/mutations in resection in combination with depletion/KO/mutation of RNaseH2 (to increase rNTPs), SETX (to increase R-loops) or OGG1 (to increase 8-oxo-G). Alternatively, for 8-Oxo-G an option would be to treat the cells with an oxidizing agent. There are several easy-to-use resection tests available both in yeast and humans, and any of them would be valid. If this observations are biologically relevant, one should expect that Exo1 is particularly important in a context of increased rNTPs whereas a boost on 8-oxo-G levels will make DNA2/BLM more essential.

Additionally, I would like to know the effect of an embedded RNA stretch on EXO1 stimulation. Such substrate has been used to test DNA2/BLM in Figure 4, but not EXO1. Considering that outside a replication context most miss incorporated rNTPs would appear in such kind of configuration, I think it is essential to test it.

REFEREE #1:

1. The Referee noted that ours is a very neat paper demonstrating how long-range DNA end resection pathways respond differentially to non-canonical structures, and that it may explain the need for multiple resection pathways. The main concern of the Referee is related to physiological relevance and he/she suggested that we test the effects of RNase H, AP endonuclease or oxidative agents on the BLM-DNA2 and EXO1 pathways to resection, by scoring for total RPA signal.

Our response: We are grateful to the Referee for praising our study. The Referee's point about biological relevance is very well taken. Accordingly, we have followed the advice of the Referee by monitoring the formation of RPA foci upon the irradiation of U2OS cells with OGG1, APE1, or RNase H2A depletion in combination with EXO1 or DNA2 knockdown. The results reveal that depletion of OGG1 or APE1 causes a greater dependency on DNA2 for resection, which is in congruence with our biochemical data showing that 8-oxo-G residues specifically impede EXO1 but enhance resection by BLM-DNA2, while AP sites stimulate unwinding by BLM and resection by BLM-DNA2. Cells with depletion of RNase H2A exhibit an overall attenuation of resection. Our interpretation is that R-loop-associated RNA:DNA hybrids, which accumulate in the absence of RNase H2A, inhibit resection by both EXO1 and BLM-DNA2 as we have observed *in vitro* (Figure 2A and B, Figure 3C). Overall, the new cell-based data attest to the physiological relevance of our biochemical findings. These new biological data are presented in Figure 6 and Figure S3 and discussed (page 9) in the revised manuscript.

2. The Referee asked whether the preference for AP sites by BLM-DNA2 occurs under reaction conditions when the concentration of Mg²⁺ is greater than that of ATP, as duplex destabilization is favored when there is no free magnesium.

Our response: Thank you for making this point. These experiments (Figure 5) were performed under conditions where the concentration of magnesium (2 mM) is greater than that of ATP (1 mM).

3. The Referee pointed out that while Dna2 was reported to be stimulated by RNA at the 5' end by the Seo laboratory (Bae et al., JBC, 2000), a different result was reported for *Pyrococcus* Dna2 (Higashibata et al., JBC, 2003). The Referee questioned what the effect of terminal RNA would be with human DNA2 using the experimental design in Figure 4.

Response: Following the Referee's advice, we have now tested a substrate with 4 nt of RNA at the end of the 5' flap. The results (Figure S2D) reveal that human DNA2 acts on this substrate with the same efficiency as it does a 5' flap with only DNA. Thus, human DNA2 seems to behave like the *Pyrococcus* enzyme. This property of human DNA2 is now mentioned and the two relevant references are cited (page 6).

REFEREE #2:

1. The Referee pointed out that Gravel et al. (2008) first discussed the possibility that the two long-range end resection machineries may each be suited to different physiological conditions,

such as during different cell cycle phases and DNA ends with base damage. The Referee acknowledged that ours is a serious attempt to tackle this question by conducting a thorough in vitro analysis of the specificity of human EXO1 and BLM-DNA2 resection machineries towards chimeric substrates containing either ribonucleotides (rNTs) or damaged bases, such as 8-oxoguanine (OG) or apurinic/apyrimidinic (AP) sites.

While finding the biochemical data and analysis to be clean and convincing and the manuscript well-written, the Referee had concerns regarding whether the observed activities reflect the physiological functions of EXO1 and BLM-DNA2. Specifically, the Referee noted that cells are equipped with DNA repair systems detecting and removing ribonucleotides via ribonucleotide excision repair (RER) and 8-oxo-G and AP sites by base excision repair (BER). BER and RER could cooperate with the HR pathway to remove these obstacles, and they may act in redundant fashion with either of the two long-range resection machineries. The Referee also noted EXO1 could substitute for FEN1 during RER with reasonable efficiency (Sparks et al.).

Our response: We are pleased that the Referee found our biochemical data and analysis to be clean and convincing and the manuscript well-written. The concern of Referee #2 resonates with that raised by the other two referees and is very well taken. As detailed in our response to Point #1 of Referee #1, we have expended considerable effort to conduct biological experiments involving the simultaneous depletion of OGG1, APE1, or RNase H2A together with either EXO1 or DNA2. The results, shown in Figure 6 and Figure S3, provide the requisite biological context to our biochemical findings revealing the unique behavior of EXO1 and BLM-DNA2 at DNA ends with non-canonical nucleotides.

We now cite and discuss the Gravel et al. 2008 paper that entertains the possibility that EXO1 and BLM-DNA2 may each be suited to different physiological settings of DNA end resection (page 7).

Regarding the Reviewer's point that BER may help to clear base damage in regions near DSBs, our new data showing that OGG1 or APE1 knockdown increases the dependence of resection on the DNA2-dependent pathway help strengthen our original conclusion (based on the in vitro data) that the HR machinery is equipped to handle these base lesions. We are grateful to the Referee for his/her insight and now thoroughly discuss this point in the Discussion (page 9).

Our new finding that RNase H2A knockdown leads to an overall attenuation of resection (Figure 6C) and that further knockdown of EXO1 or DNA2 has a modest additional effect also helps address the Referee's point. Our interpretation is that RNase H2A knockdown leads to accumulation of RNA:DNA hybrids that slow resection by either pathway, as we have shown in vitro in Figure 2 and Figure 3C. This is consistent with previous reports that the cellular consequences of loss of RNase H2 are primarily due to R-loop accumulation and not RER deficiency. We have added a series of citations (page 9) to address this point, including the Sparks et al. 2012 paper referred to by the Reviewer.

2. The Referee mentioned that the demonstration of a 5'-3' RNase H-like activity of EXO1 is not original (Qiu et al., 1999), and likely reflects its physiological function during Okazaki fragment processing (as a back-up for Rad27/FEN1; Kahli et al., 2019). Moreover, the Referee noted that

the DNA/RNA substrates and assays are not convenient to distinguish between ‘resection of replication-associated DSB ends’ or ‘Okazaki fragment processing’ by EXO1, with the same criticism being applicable to assays involving BLM-DNA2. The Referee also found the title of our manuscript highly speculative.

Our response: We have added references to the 1999 Qiu paper and the 2012 Kahli et al. paper. Our substrates to assess resection of a replication-associated DSB containing an unprocessed Okazaki fragment (Figure 1C) resemble those used by Qiu et al. to study Okazaki fragment processing. The published study used a limited number of end configurations, however, and did not pick up the robust stimulation of EXO1 by 5’ RNA that we have documented here. We note that there is no strong evidence for a primary role of EXO1 in Okazaki fragment processing under normal physiological conditions in which FEN-1, DNA2, and RNase H2 are all present. That overexpression of human EXO1 suppresses the temperature sensitivity of a *rad27* mutant yeast strain (Qiu et al. 1999) could reflect a backup role of EXO1 in processing Okazaki fragments.

Overall, we believe that together with the newly added cell-based data (Figure 6 and Figure S3), our study furnishes evidence for a differential response of the EXO1 and DNA2-dependent resection pathways to lesion context, and provides a mechanistic basis for understanding why multiple long range resection pathways have been retained throughout millennia of evolution. Given the above, we feel that the title of the manuscript does accurately reflect the implications of our findings.

3. The Referee noted that experimental data showing EXO1-dependent processing of substrates containing short internal RNA is missing, even though it is entertained in the model presented in Figure 6D.

Our response: Thank you for making the point. We have now tested this substrate (also requested by Reviewer 3) and the new data are shown in Figure 2D. Surprisingly, we find that an internal stretch of 4 nt of RNA causes EXO1 to stall, unlike 5’ terminal RNA which stimulates EXO1. The new data are discussed on pages 7 and 8. The reason why RNA at the 5’ terminus activates EXO1 but internal RNA causes it to stall is unclear, and will require structural studies which are beyond the scope of the current manuscript.

4. The Referee pointed out that DNA duplexes containing AP sites and 8-oxo-G residues have reduced thermal stability (Vesnaver et al., 1989; Singh et al., 2011), thereby potentially facilitating BLM-mediated unwinding and/or adjacent DNA2-mediated cleavage reactions carried out in vitro.

Our response: We agree with the Referee that the increased unwinding of substrates containing AP sites is likely due at least in part to destabilization of the duplex. The observation that a DNA mismatch also induces greater unwinding by BLM is consistent with this interpretation (Figure 5C). However, we note that 8-oxo-G has little effect on unwinding by BLM (Figure 5C), which suggests that features other than decreased thermal stability of DNA may also be at play. In any case, our new cell-based experiments showing increased dependence on the DNA2-dependent pathway for resection following depletion of OGG1 and APE1 (Figure 6A and B, respectively)

provide evidence that the stimulation of the BLM-DNA2 pathway we observed in vitro is physiologically relevant. We have added a paragraph to the Discussion (page 8) addressing this point and have incorporated the references that the Referee suggested.

5. The Referee noted that detailed information regarding the substrates used was missing from the manuscript.

Our response: A table listing all the oligonucleotides used to make each substrate was inadvertently left out of the original submission and has been added to the revised version as Supplemental Table 1. We apologize sincerely for this oversight.

6. The Referee pointed out that the length of unprocessed substrates in Figure 1B (a dsDNA substrate with a nick in the middle) cannot be 30 nt because the substrates used in other panels of this figure are 30 nt long in total.

Our response: The labeling in Figure 1B is correct. This substrate contains two 35-nt oligos annealed to a 70-mer. It was necessary to make this substrate longer than some of the other ones we used in the paper in order to ensure that the two shorter (top) oligos would anneal reliably to the longer (bottom) one. The details of this substrate have been clarified in the figure legend and the oligonucleotides used in substrate construction are listed in Supplemental Table 1.

7. The Referee suggested blocking the 5' end of the unlabeled strand by adding a short 5' overhang to prevent EXO1 from digesting this strand.

Our Response: We have tested the effect of 5' terminal RNA on substrates to which we have added a 4 nt 5' overhang as suggested by the Referee. The new data, shown in Figure S1B, show that 5' terminal RNA stimulates EXO1 activity on these substrates like it does on the blunt ended substrates.

8. The Referee suggested that the models shown in Figure 1 of the original submission did not add substantial value to the paper.

Our response: We thank the Referee for this suggestion and have removed this figure from the revised manuscript.

REFEREE #3

1. The Referee noted that our findings are interesting and that they might explain why eukaryotic cells have maintained different nucleases in DNA end resection. Like the other two referees, the Referee specifically suggested knocking down RNase H2 to increase rNTPs, SETX to increase R-loops, and OGG1 to increase 8-oxo-G residues and testing their effect on resection.

Our response: We thank the Referee for his/her enthusiasm and for the suggestion of experiments. As we have detailed in our response to similar points of Referee #1 and Referee #2, we have conducted several cell-based experiments to biologically validate our biochemical findings. The results are shown in Figure 6 and Figure S3 of the revised manuscript.

2. The Referee asked how an embedded RNA stretch would affect EXO1.

Our response: We have now tested this substrate (also requested by Referee 2) and the new data are shown in Figure 1D. Surprisingly, we find that an internal 4-nt stretch of RNA causes EXO1 to stall, unlike 5' terminal RNA which stimulates EXO1. The new data are discussed on page 8. Why RNA at the 5' terminus activates EXO1 but internal RNA causes it to stall is unclear, and will require structural studies which are beyond the scope of the current manuscript.

REVIEWERS' COMMENTS:

Reviewer #1 (Remarks to the Author):

I thank the authors for their effort on improving the manuscript; in particular the added cellular experiments very nicely support the biochemistry. This paper certainly sheds light on why do cells possess two pathways for long-range resection. I am happy to support its acceptance in Nature Communications.

Petr Cejka

Reviewer #2 (Remarks to the Author):

I congratulate the authors for addressing all comments to my complete satisfaction and recommend acceptance of the revised manuscript for publication after revising the following two issues:

1. The authors may want to add a sentence on page 7 explaining why depletion of OGG1 results in a 2-fold increase the number of IR-induced RPA foci (Figure 6A)? Has this phenotype observed in other studies (reference)? Is this perhaps due to increased DNA breakage (potentiation of IR)? It would have been of considerable value to include gamma-H2AX foci analysis in these experiments.
2. I suggest to state IR dose (6 Gy) and time point of RPA foci analysis (3 hours after IR) in Figure Legend 6 for better readability.

Reviewer #3 (Remarks to the Author):

The authors have answered all my concerns satisfactorily, thus I am glad to support its acceptance in Nat Comm.

Reviewer #2 (Remarks to the Author):

I congratulate the authors for addressing all comments to my complete satisfaction and recommend acceptance of the revised manuscript for publication after revising the following two issues:

1. The authors may want to add a sentence on page 7 explaining why depletion of OGG1 results in a 2-fold increase the number of IR-induced RPA foci (Figure 6A)? Has this phenotype observed in other studies (reference)? Is this perhaps due to increased DNA breakage (potentiation of IR)? It would have been of considerable value to include gamma-H2AX foci analysis in these experiments.

We thank the reviewer for this suggestion. We have added a sentence on page 7 speculating on the reason for the increased RPA foci observed in Figure 6A.

2. I suggest to state IR dose (6 Gy) and time point of RPA foci analysis (3 hours after IR) in Figure Legend 6 for better readability.

We have added this information to the Figure 6 legend.